# SE(2)-Equivariant Pushing Dynamics Models for Tabletop Object Manipulations

**Seungyeon Kim**[1]    **Byeongdo Lim**[1]    **Yonghyeon Lee**[1]    **Frank Chongwoo Park**[1,2]
[1]Seoul National University, [2]Saige Research
{ksy, bdlim, yhlee}@robotics.snu.ac.kr, fcp@snu.ac.kr

**Abstract:** For tabletop object manipulation tasks, learning an accurate pushing dynamics model, which predicts the objects' motions when a robot pushes an object, is very important. In this work, we claim that an ideal pushing dynamics model should have the SE(2)-*equivariance* property, i.e., if tabletop objects' poses and pushing action are transformed by some same planar rigid-body transformation, then the resulting motion should also be the result of the same transformation. Existing state-of-the-art data-driven approaches do not have this equivariance property, resulting in less-than-desirable learning performances. In this paper, we propose a new neural network architecture that by construction has the above equivariance property. Through extensive empirical validations, we show that the proposed model shows significantly improved learning performances over the existing methods. Also, we verify that our pushing dynamics model can be used for various downstream pushing manipulation tasks such as the object moving, singulation, and grasping in both simulation and real robot experiments. Code is available at https://github.com/seungyeon-k/SQPDNet-public.

**Keywords:** Pushing dynamics learning, Pushing manipulation, Symmetry and Equivariance

## 1   Introduction

Robotic visual pushing manipulation – by visual manipulation, we mean that only visual observations (e.g., depth camera) are available – in cluttered environments including unseen objects is an important yet challenging manipulation skill that allows a robot to interact with and change its environment to be suitable for performing downstream tasks. For example, pushing manipulation techniques have been used to move tabletop objects graspable [1, 2, 3, 4], rearrange multiple objects for sorting [5, 6, 7, 8], and find a target object occluded by the other objects [9, 10].

We consider model-based approaches for the pushing manipulations that consist of the following two components: (i) to construct a pushing dynamics model which predicts the motions of the objects after a robot performs a pushing action to the environment and (ii) to find an optimal sequence of pushing actions that achieves the goal given a predesigned task criteria [11, 12]. Our primary focus is the first step which is to develop an accurate *visual pushing dynamics model* that takes a visual observation as an input. Analytic approaches that precisely model the physical interactions [13, 14, 15, 16] cannot be used since we are given unseen objects with only vision data.

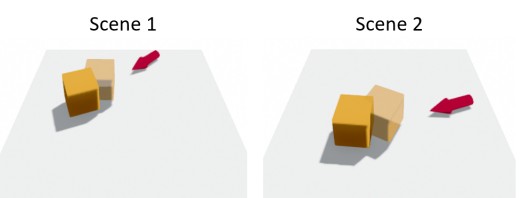

Figure 1: The box object and pushing vector in *Scene 1* are transformed by some same planar rigid-body transformation as those in *Scene 2*. An ideal pushing dynamics model should be SE(2)-equivariant, i.e., the resulting motion in *Scene 2* is a transformation of that in *Scene 1*.

Recently, there has been considerable interest in data-driven methods for *learning* pushing dynamics models [17, 18, 19, 20, 21, 22], but their generalization performances are still far-less-than-satisfying. We claim that one of the important reasons behind this is that neural network models

6th Conference on Robot Learning (CoRL 2022), Auckland, New Zealand.

used in existing approaches lack considering the symmetry of the physical systems, and more precisely, *equivariance*. For example, suppose a model is trained with an experience where a robot pushes a box object into a red arrow direction as shown in Figure 1 (*Scene 1*). And consider a new situation where the same box object is located at a different pose and the robot pushes the object in the same relative direction as shown in Figure 1 (*Scene 2*). At an intuitive level, a good model should be able to easily generalize to this type of new situation, where tabletop objects are only translated or rotated along the $z$-axis. In more technical terms, the pushing dynamics model needs to be *equivariant* to the $\mathrm{SE}(2)$ transformation.

In this paper, we define the $\mathrm{SE}(2)$-equivariant pushing dynamics model and deliberately design a neural network architecture that by construction has the equivariance property. The core idea to make the model equivariant is to properly transform the coordinates of the pushing action and the objects' poses as needed; details are elaborated in Section 2. This construction naturally captures the symmetry of the physical systems and significantly improves the generalization performances.

To employ the proposed equivariant pushing dynamics model in environments with only vision data and unseen objects, we need an additional module that can recognize the objects' shapes and poses. In this work, we represent 3d objects' shapes by using the shape class called the *superquadrics*, which can express diverse shapes ranging from boxes, cylinders, and ellipsoids to other complex symmetric shapes. We train the recognition network that predicts the objects' shapes with superquadrics by adopting an idea from [23]. We call our superquadric object representation-based pushing dynamics model a *SuperQuadric Pushing Dynamics Network (SQPD-Net)*.

Experiments and benchmark comparisons against the existing state-of-the-art methods confirm that our dynamics model achieves the highest performance in predicting objects' motions after pushing action. In addition, we validate the effectiveness of our model by using it for model-based optimal controls for various pushing manipulation tasks in both simulation and real-world experiments.

## 2    SE(2)-Equivariant Pushing Dynamics Models

In this section, we develop a neural network architecture specialized to learn a $\mathrm{SE}(2)$-equivariant pushing dynamics model. We assume that multiple rigid-body objects are placed on the table whose surface is assumed to be flat and orthogonal to the gravity direction, and the robot interacts with the objects by pushing manipulations. Each object is represented by a pose parameter $\mathbf{T} \in \mathrm{SE}(3)$ ($4 \times 4$ matrix representation) and shape parameter $\mathbf{q}$, where the pose parameter is described with respect to some global fixed frame and the shape parameter is a vector. And the pushing action is defined as a tuple $(\mathbf{p}, \mathbf{v})$ where the tip of the end-effector moves from the position $\mathbf{p} \in \mathbb{R}^3$ to $\mathbf{p} + \mathbf{v} \in \mathbb{R}^3$. As the tip of the end-effector moves, the robot can have contact with environments, pushes objects, and changes the poses of the objects.

Further, we assume there are maximally $M$ rigid-body objects on the table that have the parameters $\{(\mathbf{T}_i, \mathbf{q}_i)\}_{i=1}^N$ for $N \leq M$. We consider a discrete-time pushing dynamics model $f$ that outputs the object's transformed poses $\{\mathbf{T}'_i\}_{i=1}^N$ when a pushing action $(\mathbf{p}, \mathbf{v})$ is applied, i.e., $\{\mathbf{T}'_i\}_{i=1}^N = f(\{(\mathbf{T}_i, \mathbf{q}_i)\}_{i=1}^N, (\mathbf{p}, \mathbf{v}))$, where $N$ can vary as long as $N \leq M$. Assuming the gravity direction is the $z$-axis, we first give a precise definition of the $\mathrm{SE}(2)$-equivariant pushing dynamics model:

**Definition 1** *A pushing dynamics model $f$ is* $\mathrm{SE}(2)$-*equivariant if*

$$\{\mathbf{C}\mathbf{T}'_i\}_{i=1}^N = f(\{(\mathbf{C}\mathbf{T}_i, \mathbf{q}_i)\}_{i=1}^N, (\mathbf{Rot}(\hat{\mathbf{z}}, \theta)\mathbf{p} + \mathbf{t_{xy}}, \mathbf{Rot}(\hat{\mathbf{z}}, \theta)\mathbf{v})) \tag{1}$$

*for all object numbers $N \leq M$ and rigid-body transformations $\mathbf{C}$ that have the following form*

$$\mathbf{C} = \begin{bmatrix} \mathbf{Rot}(\hat{\mathbf{z}}, \theta) & \mathbf{t_{xy}} \\ 0 & 1 \end{bmatrix}, \tag{2}$$

*where $\mathbf{Rot}(\hat{\mathbf{z}}, \theta)$ is a $3 \times 3$ rotation matrix for rotations around $z$-axis and $\mathbf{t_{xy}} = (t_x, t_y, 0) \in \mathbb{R}^3$.*

To build a $\mathrm{SE}(2)$-equivariant neural network architecture, we first introduce an object pose decomposition method that decomposes an object pose $\mathbf{T}_i \in \mathrm{SE}(3)$ to a pose projected to the table surface denoted by $\mathbf{C}_i \in \mathrm{SE}(3)$ and the relative rigid-body transformation $\mathbf{U}_i \in \mathrm{SE}(3)$ such that $\mathbf{T}_i = \mathbf{C}_i \mathbf{U}_i$.

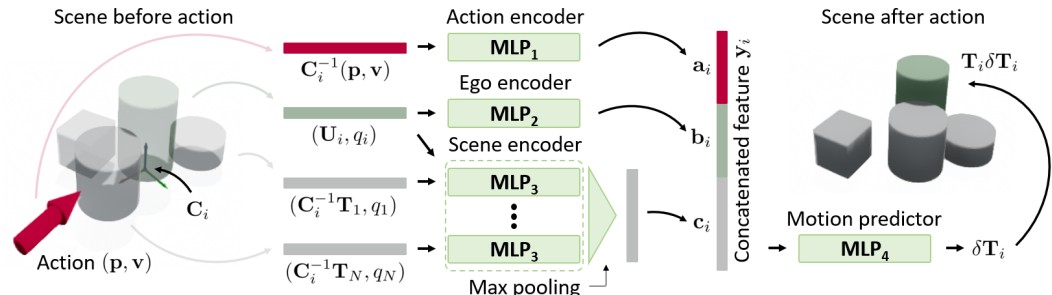

Figure 2: SE(2)-equivariant pushing dynamics neural network architecture for an $i$-th object, $f_i$.

**Object Pose Decomposition.** Given an object pose $\mathbf{T} \in \mathrm{SE}(3)$, we decompose it to two $4 \times 4$ matrices $\mathbf{C}, \mathbf{U} \in \mathrm{SE}(3)$ as visualized in Figure 3. First, $\mathbf{C}$ is defined by projecting $\mathbf{T}$ to the table surface, which has the form in equation (2). And secondly, $\mathbf{U}$ is defined as $\mathbf{C}^{-1}\mathbf{T}$. More details are in Appendix B.

Now, we explain our network architecture for the pushing dynamics model $f$; overall architecture is described in Figure 2. The model $f$ is defined as $\{f_i\}_{i=1}^N$ where each $f_i$ outputs the $i$-th object's transformed pose, i.e., $\mathbf{T}_i' = f_i(\{(\mathbf{T}_j, \mathbf{q}_j)\}_{j=1}^N, (\mathbf{p}, \mathbf{v}))$. For $f_i$, we first decompose the $i$-th object pose $\mathbf{T}_i = \mathbf{C}_i \mathbf{U}_i$ and transform the other objects' poses (including itself) and pushing action as follows: (i) $\mathbf{T}_j \mapsto \mathbf{C}_i^{-1}\mathbf{T}_j$ for $j = 1, \cdots, N$ and (ii) $(\mathbf{p}, \mathbf{v}) \mapsto \mathbf{C}_i^{-1}(\mathbf{p}, \mathbf{v}) := (\mathbf{R}_i^T \mathbf{p} - \mathbf{R}_i^T \mathbf{t}_i, \mathbf{R}_i^T \mathbf{v})$ where $\mathbf{R}_i$ and $\mathbf{t}_i$ are rotation matrix and translation vector parts of $\mathbf{C}_i$. Then, three different multi-layer

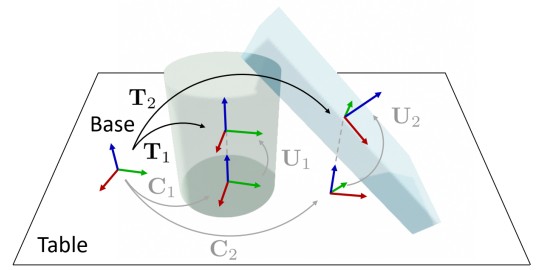

Figure 3: Object Pose Decomposition.

perceptron (MLP) networks are used to extract $\mathrm{SE}(2)$-invariant feature vectors: (i) the $\mathrm{MLP}_1$ takes the transformed action $\mathbf{C}_i^{-1}(\mathbf{p}, \mathbf{v})$ and outputs a feature vector $\mathbf{a}_i$, (ii) the $\mathrm{MLP}_2$ takes the $i$-th object's parameter $(\mathbf{U}_i, \mathbf{q}_i)$ and outputs a feature vector $\mathbf{b}_i$, and (iii) the $\mathrm{MLP}_3$ takes the transformed object's parameters $(\mathbf{C}_i^{-1}\mathbf{T}_j, \mathbf{q}_j)$ and outputs a feature vector $\mathbf{c}_i^j$ for all $j = 1, \cdots, N$ and then these output vectors pass through some permutation invariant function $h$ as $\mathbf{c}_i = h(\mathbf{c}_i^1, \cdots, \mathbf{c}_i^N)$ such as the element-wise max pooling. These feature vectors are concatenated as $\mathbf{y}_i = (\mathbf{a}_i, \mathbf{b}_i, \mathbf{c}_i)$, and we have the last layer $\mathrm{MLP}_4$ that takes $\mathbf{y}_i$ and outputs $\delta \mathbf{T}_i \in \mathrm{SE}(3)$. We note that these MLP layers are shared across all $i = 1, \cdots, N$. Then, the transformed poses are defined as $\mathbf{T}_i' = \mathbf{T}_i \delta \mathbf{T}_i$ for all $i = 1, \cdots, N$. As a result, this dynamics model is $\mathrm{SE}(2)$-equivariant by construction; the proof is in Appendix B.

**Training.** Denote by $\mathbf{s} = \{(\mathbf{T}_i, \mathbf{q}_i)\}_{i=1}^N$ for some $N \leq M$ and $\mathbf{a} = (\mathbf{p}, \mathbf{v})$. In this paper, we train the pushing dynamics model given a set of 3-tuples $\{(\mathbf{s}, \mathbf{a}, \{\mathbf{T}_i'\}_{i=1}^N)_k\}_{k=1}^K$ where $\mathbf{T}_i'$ is the next pose of the $i$-th object. The loss function $\mathcal{L}$ is defined by comparing the ground-truth next poses $\{\mathbf{T}_i'\}_{i=1}^N$ and the predicted poses $\{\hat{\mathbf{T}}_i'\}_{i=1}^N = f(\mathbf{s}, \mathbf{a})$ as follows:

$$\mathcal{L}(f) = \sum_{i=1}^N \left( \|\mathbf{t}_i' - \hat{\mathbf{t}}_i'\|_2^2 + \alpha \cdot d_{\mathrm{SO}(3)}^2(\mathbf{I}_3, \mathbf{R}_i'^{-1}\hat{\mathbf{R}}_i') \right), \tag{3}$$

where $d_{\mathrm{SO}(3)}$ is a distance measure between two rotation matrices, $\mathbf{I}_3$ is $3 \times 3$ identity matrix, $\mathbf{R}_i', \hat{\mathbf{R}}_i'$ and $\mathbf{t}_i', \hat{\mathbf{t}}_i'$ are rotation matrices and translation vectors parts of $\mathbf{T}_i', \hat{\mathbf{T}}_i'$, respectively, and $\alpha$ is a weighting parameter (for our later experiments we set $\alpha$ to 0.1). The details about the used distance measure are in Appendix C.

# 3 Object Recognition-based Pushing Manipulations

If we have known objects and can easily estimate the poses of the objects, then it is straightforward to use the learned pushing dynamics model for pushing manipulations. However, for unseen objects, we first need to recognize the objects' shapes and poses. Therefore, our overall framework consists of the following two steps: (i) to recognize objects' shapes and poses and (ii) to push objects by using the learned pushing dynamics model and pre-designed task criteria, of which details are explained in the following subsections.

## 3.1 Object Shape and Pose Recognition via Superquadrics

We propose to use implicit functions to represent 3d objects' shapes. In general, an implicit object surface representation is defined by a level set of a function $S(x, y, z; \mathbf{q}, \mathbf{T}) = 0$, where $\mathbf{q}$ is a shape parameter and $\mathbf{T} \in \mathrm{SE}(3)$ is a pose parameter. In our framework, any implicit function approximation model $S(x, y, z; \mathbf{q}, \mathbf{T})$ can be used.

In this work, we employ the shape class called the superquadrics, a family of geometric shapes that resemble ellipsoids and other quadrics, which can be used to represent diverse shapes ranging from boxes, cylinders, and ellipsoids to bi-cones, octahedra, and other complex symmetric shapes. The implicit equation for a superquadric surface at $\mathbf{T} = \mathbf{I}_4$ ($\mathbf{I}_4$ is $4 \times 4$ identity matrix) has the following form:

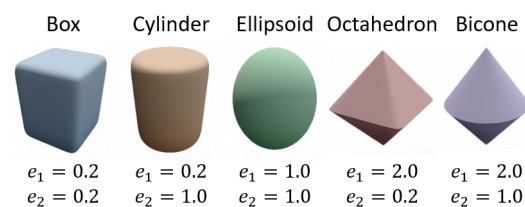

Figure 4: Examples of superquadrics.

$$S(x, y, z; \mathbf{q}, \mathbf{I}_4) = \left( \left| \frac{x}{a_1} \right|^{\frac{2}{e_2}} + \left| \frac{y}{a_2} \right|^{\frac{2}{e_2}} \right)^{\frac{e_2}{e_1}} + \left| \frac{z}{a_3} \right|^{\frac{2}{e_1}} - 1 = 0, \tag{4}$$

where $\mathbf{q} = (a_1, a_2, a_3, e_1, e_2) \in \mathbb{R}^5$ is the shape parameter. In particular, $a_1, a_2, a_3$ controls the sizes and $e_1, e_2$ controls the geometric shapes. Some examples are shown in Figure 4. At $\mathbf{T} \neq \mathbf{I}_4$, the equation $S(x, y, z; \mathbf{q}, \mathbf{T})$ can be written with the passive coordinate transformation of $(x, y, z)$ by $\mathbf{T}$, i.e., $S(x, y, z; \mathbf{q}, \mathbf{T}) = S(\mathbf{T}^{-1}(x, y, z); \mathbf{q}, \mathbf{I}_4)$; see Appendix C for details.

The object recognition problem that we address in this paper can then be posed as follows: given a visual input obtained from a depth camera that typically contains partial views of the objects, we need to predict the superquadric parameters $(\mathbf{q}, \mathbf{T})$ for each object. To bridge the gap between synthetic and real-world vision sensor data, we add noise to the visual input as done in [24, 25]. The predicted object represented by $(\mathbf{q}, \mathbf{T})$ should fit the full object, although only a partial view of the object is given as input. This problem has been recently tackled by [23], where two neural network models that take point cloud data as inputs are employed: (i) object segmentation network [26] and (ii) object full shape and pose recognition network [23]. We include details about the visual input noise, the network architectures, and the training methods of these networks in Appendix C.

We call our $\mathrm{SE}(2)$-equivariant pushing dynamics model that uses the superquadric representation a *SuperQuadric Pushing Dynamics Network (SQPD-Net)*.

## 3.2 Model-based Pushing Manipulations

Given a visual observation of tabletop objects as a point cloud which we denote by $\mathbf{o}$, our goal is to find a sequence of robot pushing actions $(\mathbf{a}_1, \mathbf{a}_2, \cdots, \mathbf{a}_T)$ that changes the environment for some given task. In this section, we assume that we are given (i) a recognition module $R$ that outputs the objects' poses and shapes, i.e. $R(\mathbf{o}_t) = \mathbf{s}_t$ (throughout, we denote by $\mathbf{s}_t = \{(\mathbf{T}_{t,i}, \mathbf{q}_{t,i})\}_{i=1}^N$), and (ii) a pushing dynamics model $\mathbf{s}_{t+1} = f(\mathbf{s}_t, \mathbf{a}_t)$. Given a task-specific objective function $\mathcal{J}$, we solve the following optimal control problem:

$$\min_{\mathbf{a}_1, \cdots, \mathbf{a_T}} \mathcal{J}(\mathbf{o}_1, \mathbf{a}_1, \cdots, \mathbf{a}_T) = \sum_{t=1}^{T} r(\mathbf{s}_t, \mathbf{a}_t) + q(\mathbf{s}_{T+1}) \ \text{ s.t. } \ \mathbf{s}_1 = R(\mathbf{o}_1), \ \ \mathbf{s}_{t+1} = f(\mathbf{s}_t, \mathbf{a}_t). \tag{5}$$

For tasks we focus in this paper, we set $r(\mathbf{s}_t, \mathbf{a}_t) = 0$ and only use a terminal cost function $q(\mathbf{s}_{T+1})$. We use the sampling-based MPCs [12] (implementation details are in Appendix D). Below, we

introduce three terminal cost functions for the following pushing manipulation tasks: (i) moving, (ii) singulation, and (iii) grasping. We denote the translation vector and rotation matrix parts of the transformation matrix $\mathbf{T}_{(\cdot)}$ as $\mathbf{t}_{(\cdot)}, \mathbf{R}_{(\cdot)}$, respectively.

**Moving** is a task to move objects to their desired poses. The desired poses are given as $\{\mathbf{T}_{d,i}\}_{i=1}^{N}$, then we define a terminal cost function as

$$q(\mathbf{s}_{T+1}) = \sum_{i=1}^{N} \left( \|\mathbf{t}_{T+1,i} - \mathbf{t}_{d,i}\|_2^2 + \beta \cdot d_{\mathrm{SO(3)}}(\mathbf{I}_3, \mathbf{R}_{d,i}^{-1}\mathbf{R}_{T+1,i}) \right). \tag{6}$$

**Singulation** is a task to separate objects by more than a certain distance $\tau$. We define a terminal cost function as

$$q(\mathbf{s}_{T+1}) = - \min_{\{(i,j)\in\{1,\cdots,N\}|i>j\}} \left( \min(\|\mathbf{t}_{T+1,i} - \mathbf{t}_{T+1,j}\| - \tau, 0) \right). \tag{7}$$

**Grasping** is a task to make a target object graspable. Given a target object index $i$, we generate candidate grasp poses for the recognized target object as shown in Figure 5 and check collisions with the environment and the other recognized objects; green grasp poses are collision-free and red poses are not. The terminal cost $q(\mathbf{s}_{T+1})$ is defined to be 0 if at least one collision-free grasp pose exists and 1 otherwise. Further details are provided in Appendix C.

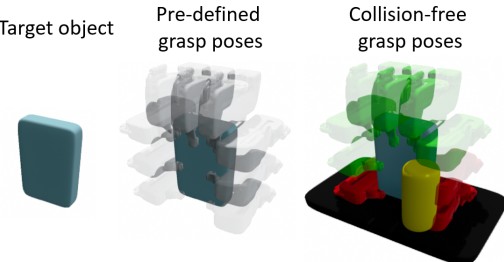

Figure 5: Sampling-based grasping criteria.

## 4    Experiments

In this section, we empirically show that (i) our proposed pushing dynamics model, the SQPD-Net, outperforms the existing state-of-the-art data-driven pushing dynamics models, and (ii) our SQPD-Net can be used for various downstream pushing manipulation tasks, e.g., object moving, singulation, and grasping.

**Environment.** We use the 7-dof Franka Emika Panda robot with a parallel-jaw gripper and an Azure Kinect DK camera sensor mounted on the gripper. The raw input visual observation is a depth image, which is then pre-processed to other 3d representations (e.g., point cloud) as needed.

**Pushing Manipulation Dataset.** To train pushing dynamics models, we generate a pushing manipulation dataset in simulation (Pybullet). Throughout our experiments, we use cylinder-shaped and cube-shaped objects of various sizes, and one scene contains less than 5 objects. To execute an action $(\mathbf{p}, \mathbf{v}) \in \mathbb{R}^6$, (i) the robot first moves so that the gripper's tip is placed at $\mathbf{p}$ and its orientation is set as visualized in Figure 6, and then (ii) the robot moves in a way that the gripper's tip moves to $\mathbf{p} + \mathbf{v}$ with a fixed orientation. We generate the pushing manipulation dataset as follows: (i) we place random objects at random poses in the workspace, (ii) we sample an action $(\mathbf{p}, \mathbf{v}) \in \mathbb{R}^6$, where $\mathbf{p}$ is sampled near one randomly selected object and $\mathbf{v}$ directs the center of the object, and (iii) we execute the robot pushing action. In this process, we note that the gripper's other parts than the tip can also make contact with the environment. More details are included in Appendix D.

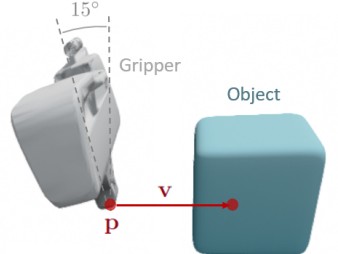

Figure 6: Execution of a pushing action.

**Baseline Methods.** We compare our SQPD-Net with the following baseline methods: *2DFlow* and *SE3-Net* adopted from [17], *SE3Pose-Net* adopted from [18], and *3DFlow* and *DSR-Net* adopted from [20]. The 2DFlow, SE3-Net, and SE3Pose-Net take an organized point cloud as a visual input and predict the flow vectors of the points. The 3DFlow and DSR-Net take a voxelized truncated signed distance field (TSDF) as a visual input and predict the voxel flow. Our SQPD-Net takes the estimated objects' poses and superquadric shape parameters as input and predicts the objects' next

poses. While, in the existing approaches, the models directly predict motions from the pre-processed raw visual observations, our model consists of two modules: (i) a pre-trained recognition network $R$ that predicts objects' poses and shape parameters and (ii) the SQPD-Net that predicts the objects' next poses. We denote these two networks together by *R-SQPD-Net*. For the comparison purpose, we also test the case where the ground-truth objects' poses and shape parameters are used as an input for the SQPD-Net and denote it by *GT-SQPD-Net*.

**Evaluation Metrics.** Throughout, we use two types of evaluation metrics for the learned pushing dynamics models: (i) flow error (the lower the better) and (ii) mask intersection over union (mask IoU, the higher the better). First of all, we consider the visible and full flow errors. The visible flow error is the root mean squared error (RMSE) between the ground-truth flows and predicted flows of the points on the visible surface of the objects, while the full flow error is the RMSE computed with all points from the objects' volumes. Second, we consider the 2D and 3D mask IoUs. The 2D mask IoU is computed by using the depth images and thus only visible surfaces are taken into consideration. On the other hand, the 3D mask IoU is computed with the complete 3D occupancy grid. The full flow error and mask IoU cannot be computed in 2DFlow, SE3-Net, and SE3Pose-Net, because they do not estimate the complete objects' shapes as an intermediate step of the prediction of the pushing dynamics.

## 4.1 Pushing Dynamics Learning

In this section, we first empirically verify the equivariance property of our method and show the performance advantages of ours over the existing methods.

**Equivariance Study.** For the purpose of testing the equivariance of the models, we design the following experiment: we train the models with *only one* pushing manipulation data – a 3-tuple $\{o, a, o'\}$ where $o$ and $o'$ are current and next observations respectively and $a$ is pushing action – so that the models overfit the given data. Then, we compare the models' generalization capabilities with test data that are generated by applying random $SE(2)$-transformation to the data. An ideal equivariant model should produce almost zero error in the test data.

| METHOD | visible flow ($\downarrow$) |
|---|---|
| 2DFlow [17] | 4.73 |
| SE3-Net [17] | 4.73 |
| SE3Pose-Net [18] | 4.72 |
| R-SQPD-Net (ours) | 0.73 |
| GT-SQPD-Net (ours) | 0.02 |

Table 1: Test visible flow error (cm).

Table 1 shows average visible flow errors of the baseline methods and SQPD-Nets, obtained by running the above experiment multiple times with different training data (details are in Appendix D). The 3DFlow and DSR-Net are omitted in this experiment because they cannot make estimations if the transformed actions do not belong to the pre-defined discrete set of actions. The GT-SQPD-Net produces almost zero error as expected while the R-SQPD-Net produces a little error originating from the recognition error. Our SQPD-Nets are much more $SE(2)$-equivariant compared to the existing works. Figure 7 shows an example prediction result from the SE3Pose-Net and R-SQPD-Net; the blue bounding box represents

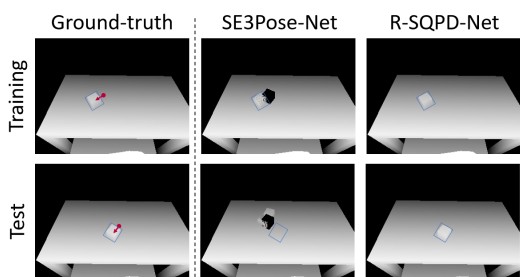

Figure 7: Depth images of prediction results. For SE3Pose-Net, after the point cloud moves, the space occupied before is colored black.

the ground-truth next pose of the object. For the test data, the SE3Pose-Net predicts a completely wrong motion. More example figures of experimental results are provided in Appendix D.

**Pushing Dynamics Learning.** We compare the learning performances of the SQPD-Nets and the baseline methods with a large-scale pushing dataset where the training/validation/test data consist of 12000, 1200, and 1200 numbers of 3-tuples ($\{o, a, o'\}$), respectively. Figure 8 shows the predicted depth images and 3D masks for an example pushing data in the test dataset. As shown in the ground-truth motions (left of Figure 8), the red, green, and gray objects are in contact with each other and these three objects move together when the red object is pushed. In this case, our R-SQPD-Net only successfully predicts the complex interactive motions of the objects. Table 2 shows

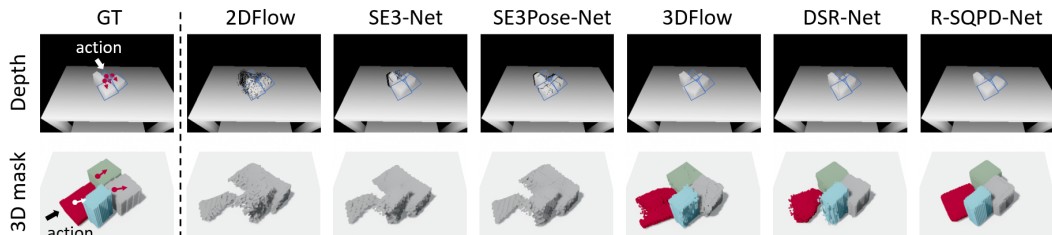

Figure 8: Depth images and 3D masks of the ground-truth next scene and predicted scenes. *Upper*: Depth images where the blue bounding boxes represent the ground-truth next poses of the green and gray objects. *Lower*: (i) (incomplete) 3D masks converted from the depth images for 2DFlow, SE3-Net, and SE3Pose-Net and (ii) predicted complete 3D masks for 3DFlow, DSR-Net, and R-SQPD-Net.

| | Known | | | | Unknown | | | |
| | Flow error (↓) | | Mask IoU (↑) | | Flow error (↓) | | Mask IoU (↑) | |
| METHOD | visible | full | 2D | 3D | visible | full | 2D | 3D |
|---|---|---|---|---|---|---|---|---|
| 2DFlow [17] | 2.179 | - | - | - | 2.180 | - | - | - |
| SE3-Net [17] | 1.631 | - | - | - | 1.701 | - | - | - |
| SE3Pose-Net [18] | 1.639 | - | - | - | 1.712 | - | - | - |
| 3DFlow [20] | 1.818 | 1.859 | 0.747 | 0.699 | 1.697 | 1.719 | 0.755 | 0.698 |
| DSR-Net [20] | 1.325 | 1.331 | 0.720 | 0.705 | 1.531 | 1.524 | 0.665 | 0.632 |
| R-SQPD-Net (ours) | **0.575** | **0.610** | **0.844** | **0.798** | **0.710** | **0.726** | **0.834** | **0.781** |
| GT-SQPD-Net (ours) | 0.519 | 0.379 | 0.903 | 0.888 | 0.638 | 0.485 | 0.888 | 0.868 |

Table 2: Evaluation metrics computed within test dataset (the unit of flow error is cm).

the evaluation metrics computed within the test data and shows that our SQPD-Nets outperform the other baseline methods by significant margins. Further experimental results with more example figures are provided in Appendix D.

## 4.2 Pushing Manipulation using R-SQPD-Net

In this section, we use the R-SQPD-Net trained in Section 4.1 and conduct the pushing manipulation tasks introduced in Section 3.2 (moving, singulation, and grasping) in both simulation and real-world. For the real-world experimental setup, we use various box- or cylinder-like objects as shown in Figure 9; the same objects are used in simulation experiments. Since we directly apply the R-SQPD-Net trained in simulation to the real physical environment, it is reasonable to ask about the sim-to-real transfer issue. In our experiments, we use slow pushing

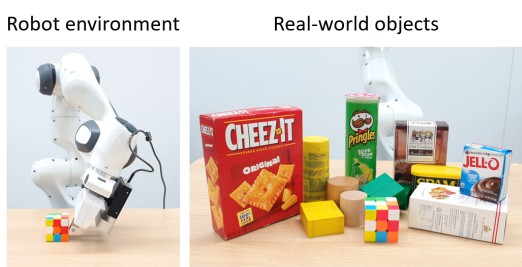

Figure 9: Real-world experimental setting.

motions to generate quasi-static movements of the objects and thus minimize the sim-to-real gap (for quasi-static object movements, the dynamical properties of the objects and environment, e.g., mass, friction coefficient, become less affective [27]).

Figure 10 shows some real-world manipulation results for various tasks. For the moving task (*first row*), we set the desired positions $\mathbf{t}_{d,i}$ as $(0.3, \mathbf{t}_{0,i,y}, \mathbf{t}_{0,i,z})$ and $\beta = 0$ in equation (6). For the singulation task (*second row*), we set $\tau = 20$ (cm) in equation (7). For the grasping tasks (*third and fourth row*), we sample about 15 to 30 candidate grasp poses for the target recognized objects. For all three examples, our approach can find a series of pushing actions that successfully perform the desired tasks. Notably, for the grasping tasks, without using ad hoc objective functions, the robot realizes how to re-configure the objects so that feasible grasp poses can be found for the target objects: (i) the robot pushes the large and flat object to the edge of the table and (ii) the robot pushes the surrounding objects to make the surrounded target object graspable.

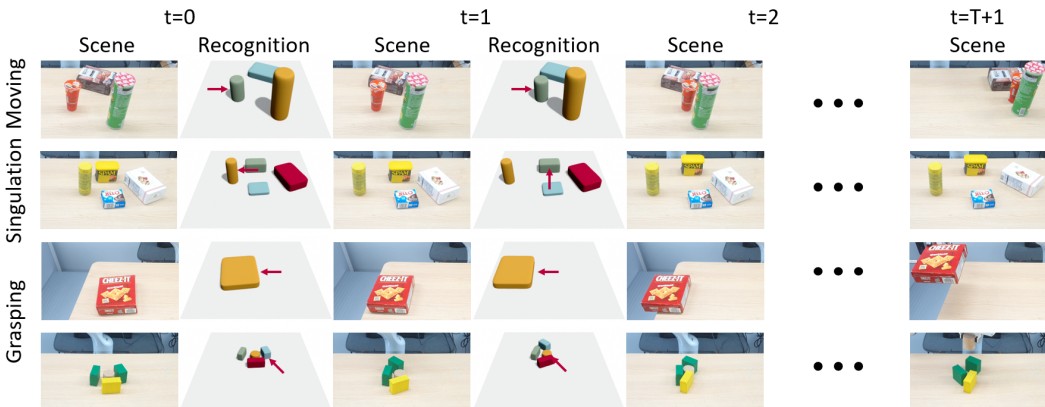

Figure 10: Real-world manipulation results using R-SQPD-Net for moving, singulation, and grasping tasks (for the *fourth row* case, the target object is the cylinder surrounded by the three cubes). The red arrow at each recognition step means the optimal pushing action.

**Failure Cases.** Table 3 shows the manipulation success rates in simulation and real-world experiments. We design 10 test scenarios for each task, of which object configurations are in Appendix D. A few failure cases occur, whose underlying reasons we observe can be roughly categorized as (i) a failure of shape recognition (simulation, real) and (ii) a sim-to-real transfer issue (real). Details can be found in Appendix D.

| TASK | Simulation | Real |
|---|---|---|
| Moving | 9/10 | 8/10 |
| Singulation | 9/10 | 8/10 |
| Grasping clutter | 4/5 | 4/5 |
| Grasping large | 4/5 | 3/5 |

Table 3: Simulation and real-world manipulation results.

### 4.3 Limitations and Future Directions

First, since SQPD-Net considers single superquadric-shaped objects, it is not trivial to apply it directly to more complex or non-convex shapes. As the researches on representing objects in multiple superquadrics progress [28, 29], extending our approach to multiple superquadric-shaped objects remains a future work. Second, the dynamics prediction task becomes challenging when pushing an object with a non-uniform mass distribution since different mass distributions will lead to different motions. In this case, if we can consistently predict the reference poses of the objects (e.g., pre-specified poses in CAD models), our $SE(2)$-equivariant model is applicable regardless of the mass distribution. Since predicting reference poses is not easy with only depth images, this is a limitation of our approach. As one possible solution, additional information such as RGB images should be utilized [30, 31]. Lastly, there could be some situations where the $SE(2)$-equivariance does not apply; in this case, our approach can be detrimental. One example is that the friction coefficients are different in different regions of the table. As a research direction to overcome this, a locally $SE(2)$-equivariant model – $SE(2)$ space is divided into several subspaces and the model is equivariant only within each subspace – can be considered.

## 5 Conclusion

This paper has proposed a $SE(2)$-equivariant pushing dynamics model. Using the superquadric representations of object shapes, we have proposed a SuperQuadric Pushing Dynamics Network (SQPD-Net). Through extensive empirical validations, we confirm that the SQPD-Net significantly outperforms the existing state-of-the-art visual pushing dynamics models. Moreover, we have verified that the SQPD-Net can be used for various pushing manipulation tasks.

**Acknowledgments**

This work was supported in part by SRRC NRF grant 2016R1A5A1938472, IITP-MSIT grant 2021-0-02068 (SNU AI Innovation Hub), IITP-MSIT grant 2022-0-00480 (Training and Inference Methods for Goal-Oriented AI Agents), SNU-AIIS, SNU-IAMD, SNU BK21+ Program in Mechanical Engineering, SNU Institute for Engineering Research, Samsung Research, and Samsung Electronics Co.,Ltd.

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
