# OpenReview forum: "SE(2)-Equivariant Pushing Dynamics Models for Tabletop Object Manipulations"
_robot-learning.org/CoRL/2022/Conference — CoRL 2022 Oral_

### Official Review · Reviewer_PwZa · 2022-07-11

**Originality:** Good
**Technical Quality:** Very Good
**Clarity Of Presentation:** Good
**Impact:** 3

**Recommendation:**

Weak Accept: I recommend accepting the paper, but will not argue for my recommendation if the majority of other reviewers have a different opinion.

**Summary:**

The paper studies the SE(2)-equivariance in robotic pushing and proposes a pushing dynamics model that is designed to be SE(2)-equivariant. Experimental results in both simulation and the real world demonstrate that 1. the proposed model achieves high-precision prediction; 2. the model can be used in various tasks.

**Issues:**

See the major issues in ‘Weaknesses’.

Minor issues:

1. line 100: $T_j \mapsto C_i^{-1}T_j$ should be $U_j \mapsto C_i^{-1}T_j$
2. Is $I_3$ in Equation 3 the identity matrix? It would be better to mention it in the text. Same applies to the $I_4$ in line 138.

**Quality Of The Limitations Section:**

Limitations are not well addressed

**Reviewer Expertise:**

4: The reviewer is confident but not absolutely certain that the evaluation is correct

**Robotics Focus:**

Sufficient demonstration on hardware

**Strengths And Weaknesses:**

Strengths:

1. The paper studies an interesting problem that utilizes SE(2)-equivariance to improve a pushing dynamics model.
2. The paper has comprehensive experimental evaluation.
3. The paper has very nice figure illustrations.

Weaknesses:

1. The paper does not have a limitation section (though there is a future direction subsection in Section 5).
2. In my opinion, the idea of performing a transformation on the input before evaluating through the network to implement equivariance is not new [1][A], though the prior works are image-based and perform transformation w.r.t. the gripper frame, but this paper is vector-based and perform transformation w.r.t. the object frame. A proper comparison would be beneficial in a related work section.
3. The paper does not discuss related works in equivariant learning and other works that implement equivariant learning in manipulation [B][C][D].


[A] Ha, Huy, and Shuran Song. "Flingbot: The unreasonable effectiveness of dynamic manipulation for cloth unfolding." *CoRL* 2021.

[B] Wang, Dian, Robin Walters, Xupeng Zhu, and Robert Platt. "Equivariant $ q $ learning in spatial action spaces." *CoRL* 2021.

[C] Wang, Dian, Robin Walters, and Robert Platt. "SO (2) Equivariant Reinforcement Learning." *ICLR* 2022.

[D] Simeonov, Anthony, Yilun Du, Andrea Tagliasacchi, Joshua B. Tenenbaum, Alberto Rodriguez, Pulkit Agrawal, and Vincent Sitzmann. "Neural descriptor fields: Se (3)-equivariant object representations for manipulation." *ICRA* 2022.

**Summary Of Recommendation:**

The idea of the paper, making use of equivariance in robotic pushing, is interesting. I believe the paper makes a good contribution, however, the paper can be improved by adding a related work section and a limitation section. Though the paper talks about some limitations in the future directions subsection in Section 5, the discussion there seems a bit superficial.

---

> ### Author Response · Authors · 2022-08-23
> **Response to Reviewer PwZa (1/1)**
>
> __Q1.__ The paper does not have a limitation section (though there is a future direction subsection in Section 5).
>
> __A1.__ Thank you for pointing out this issue. ``We now have added Section 4.3 (Limitations and Future Works) to describe the limitations of our approach more clearly.`` The future direction subsection previously in Section 5 is now included in this Section 4.3.
>
> __Q2.__ In my opinion, the idea of performing a transformation on the input before evaluating through the network to implement equivariance is not new [1][A], though the prior works are image-based and perform transformation w.r.t. the gripper frame, but this paper is vector-based and perform transformation w.r.t. the object frame. A proper comparison would be beneficial in a related work section. The paper does not discuss related works in equivariant learning and other works that implement equivariant learning in manipulation [B][C][D].
>
> __A2.__ Thank you very much for your many suggested references; we fully agree with the reviewer's comment that the review of related studies about invariance and equivariance in deep learning and robot manipulations was missing. We have accordingly ``added the related work section about invariance and equivariance in Appendix A.4 (Invariance and Equivariance in Deep Learning and Robot Manipulations).`` In this Appendix, we compare our work with these related works and clearly state at which points the proposed method is novel.
>
> __Q3.__ line 100: $\mathbf{T}_j \mapsto \mathbf{C}_i^{-1}\mathbf{T}_j$ should be $\mathbf{U}_j \mapsto \mathbf{C}_i^{-1}\mathbf{T}_j$.
>
> __A3.__ In this part, we want to denote that the pose of the $j$-th object $\mathbf{T}_j \in \mathrm{SE}(3)$ is expressed with respect to the projected pose of the $i$-th object $\mathbf{C}_i \in \mathrm{SE}(3)$. Therefore, $\mathbf{T}_j \mapsto \mathbf{C}_i^{-1}\mathbf{T}_j$ is the correct expression here.
>
> __Q4.__ Is $\mathbf{I}_3$ in Equation 3 the identity matrix? It would be better to mention it in the text. Same applies to the $\mathbf{I}_4$ in line 138.
>
> __A4.__
> Thank you for raising this point. The reviewer's comment is correct, and ``we have mentioned this in the revised paper.``

---

> > ### Comment · Reviewer_PwZa · 2022-08-26
> > **Response to Author**
> >
> > The reviewer thanks the authors for their detailed response. The revision addresses most of my concerns, and I keep my recommendation for accepting the paper.

---

> ### Author Response · Authors · 2022-08-23
> **Response to Reviewer PwZa (Revised Paper and Supplementary Material)**
>
> **Comment:**
>
> Thank you very much for your constructive feedback. In response to the many constructive suggestions we have received, we have spent the past week revising our manuscript accordingly. __The revised paper and supplementary materials are attached to this comment.__ In an attempt to answer the reviewer questions, and to better clarify and validate our contributions, we include the new additional contents in the revised manuscript as follows:
>
> * we have added Appendix E.3 (Pushing Dynamics Learning on Real-world Pushing Data), which validates the motivation and significance of our method more clearly;
> * we have added Appendix E.4 (Pushing Manipulation via Interaction);
> * we have added Section 4.3 (Limitations and Future Works) to describe the limitations of our approach more clearly;
> * we have added Appendix A.4 (Invariance and Equivariance in Deep Learning and Robot Manipulations);
> * we have added pushing manipulation demo videos in the following Youtube link: https://youtu.be/OLoAHhf7vk0;
> * we have added some more details of pushing manipulation methods in Appendix D;
> * we have fixed some typos and clarified some missing definitions.
>
> In addition to these changes, we have noted that there was a minor glitch in our data generation method; the robot exhibited irregular movements on the simulator when the robot's inverse kinematics were not solved properly, resulting in data pushing the objects irregularly. It occurred a few times during the experiment (15 data in a total of 14400 data). We have generated a new dataset and trained our model and all of the baseline models on this new dataset, then we have updated our figures and table accordingly; specifically, Tables 2, 3, Figures 8 and 10 of the manuscript, and Figures 13 and 14 of the appendix (of the revised version) are updated. These irregular movements were rare, thus none of our observations and conclusions have changed. We regret that it was a mistake that should have been carefully checked prior to submission. After rigorous verifications, we are now very confident that the results in the updated manuscript are genuine.
>
> Below we provide detailed responses to each of the reviewer comments. When referencing any major changes and the addition of new content to the revised manuscript, we have indicated those passages ``like this``.
>
> **Zip File:**
>
> /attachment/df6cb0d5f17259fa13b0573eb6f6f6ff881b2c94.zip

---

### Official Review · Reviewer_tXbi · 2022-07-31

**Originality:** Good
**Technical Quality:** Very Good
**Clarity Of Presentation:** Very Good
**Impact:** 3

**Recommendation:**

Weak Accept: I recommend accepting the paper, but will not argue for my recommendation if the majority of other reviewers have a different opinion.

**Summary:**

This paper presents an SE(2) equivariant representation of planar object dynamics which can be used to find pushing actions to drive a set of objects to a goal using planar pushing. The approach is trained with depth data to predict dynamic performance in a way that preserves geometric symmetry in planar motion. This approach is used to find pushing actions for a real robotic system and complete a wide variety of tasks. The results demonstrate how this approach outperforms previous paradigms to solve planar manipulation tasks with visual information.

**Issues:**

1. I would like to point out that equivariance is not always beneficial since different mass distributions will lead to different optimal policies and contact interactions. For instance, a square with a shifted center of mass will tend to rotate even when pushed by the middle. This is a natural limitation of this work and should be discussed with more care.
2. The usage of superquadrics limits the algorithm to convex bodies, which leaves behind a wide range of objects of interest. The authors should also discuss this limitation.
3. The authors briefly mention how they run their real-world experiments in a quasi-static fashion. How does this compare with the motion predictor? Why is this a straightforward way to close the sim-to-real gap?
4. The proposed approach vastly outperforms previous work on this problem, it would be helpful if the paper briefly explained why the previous methods tend to perform poorly on some of these tasks. Is this related to equivariance or to the object representation?

**Quality Of The Limitations Section:**

Additional details required

**Reviewer Expertise:**

5: The reviewer is absolutely certain that the evaluation is correct and very familiar with the relevant literature

**Robotics Focus:**

Sufficient demonstration on hardware

**Strengths And Weaknesses:**

Strengths:

1. Paper is easy to follow and well explained overall, along with helpful descriptive figures.
2. The approach is technically correct and
3. This method is well validated with a variety of experiments and tasks, demonstrating its benefits.

Weaknesses:

1. The paper doesn't take enough time to describe the fundamental limitations of this approach and the type of tasks where equivariance might be detrimental.
2. The limitations of superquadrics as a shape representation are not discussed.
3. The paper contains several typos and uses passive voice very frequently, which sometimes leads to confusing statements.

**Summary Of Recommendation:**

The paper is well written, presents an interesting idea, and is validated with a wide variety of experiments. The reviewer recommends acceptance. However, there are many areas of improvement when describing the limitations of the approach. More specifically, the choice of superquadrics and the impact of equivariance may lead to some fundamental limitations on object shape and mass distributions.

---

> ### Author Response · Authors · 2022-08-23
> **Response to Reviewer tXbi (2/2)**
>
> __Q3.__ The authors briefly mention how they run their real-world experiments in a quasi-static fashion. How does this compare with the motion predictor? Why is this a straightforward way to close the sim-to-real gap?
>
> __A3.__ Thank you for the great suggestions. ``We have reported the implementation of the quasi-static movement in Appendix D.1 (Details for Pushing Manipulation Data Generation).`` The trajectory of the robot pushing motion is divided into 10 via points and the end effector of the robot reaches the via points sequentially and slowly, making the object as quasi-static as possible. We did not quantitatively confirm how well the motion predictor trained on the simulator works in the real-world, but we believe we indirectly have confirmed that our motion predictor works well through real-world manipulation performance. For more information on the performance of real-world manipulation, we refer to the ``newly added experiments on a newly designed task, a challenging task that essentially requires the interaction between multi-objects, in Appendix E.4 (Pushing Manipulation via Interaction) and some real-world pushing manipulation demo videos in the following Youtube link`` (https://youtu.be/OLoAHhf7vk0). During a quasi-static push, the object only moves together when the end effector moves and stops when the end effector stops. Therefore, the motion of the object is greatly affected by the motion of the end-effector and not greatly affected by the physical properties of the object itself. In addition, to completely eliminate the sim-to-real gap, our model must be trained from real-world data. To make sure our model also works well for the real-world, ``we have added Appendix E.3 (Pushing Dynamics Learning on Real-world Pushing Data).`` We confirm that our model can also be successfully trained on a real-world dataset. We hope that these discussions and additional contents are of some help.
>
> __Q4.__ The proposed approach vastly outperforms previous work on this problem, it would be helpful if the paper briefly explained why the previous methods tend to perform poorly on some of these tasks. Is this related to equivariance or to the object representation?
>
> __A4.__ For this point, we want to refer to the experimental results in the equivariance study of Section 4.1. These experimental results suggest that our model has a higher performance than other baselines because the model is SE(2)-equivariant. We believe that SE(2)-equivariance property serves a powerful inductive bias for designing a pushing dynamics model.
>
> __Q5.__ The paper contains several typos and uses passive voice very frequently, which sometimes leads to confusing statements.
>
> __A5.__
> Thank you for raising this issue. After rigorously reviewing the manuscript, ``we have fixed some typos and modified some expressions pointed out.``

---

> ### Author Response · Authors · 2022-08-23
> **Response to Reviewer tXbi (1/2)**
>
> __Q1.__ The paper doesn't take enough time to describe the fundamental limitations of this approach and the type of tasks where equivariance might be detrimental. I would like to point out that equivariance is not always beneficial since different mass distributions will lead to different optimal policies and contact interactions. For instance, a square with a shifted center of mass will tend to rotate even when pushed by the middle. This is a natural limitation of this work and should be discussed with more care.
>
> __A1.__ Thank you for pointing out this issue. We agree with the reviewer's comment that different mass distributions will lead to different motions (e.g., a cube with a shifted center of mass). First of all, although it may already be apparent to the reviewer, we want to clarify that our model aims to guarantee SE(2)-equivariance in the situation where objects are placed in a different pose transformed by SE(2) transformation, but not aims to design equivariant models that capture the symmetry of the objects. In other words, our equivariant model considers the cases in which the objects and pushing actions are transformed by the same SE(2) transformation, not the cases in which the action pushes another symmetrical part of the object. That being said, if we can consistently predict the reference poses of the objects (e.g., pre-specified poses in CAD models), our SE(2)-equivariant model is applicable regardless of the mass distribution. Of course, predicting reference poses is not easy with only depth images, so this is still one of the limitations of our approach. In this case, we have to utilize additional information such as RGB images [1,2]. ``We have added these discussions about these limitations to Section 4.3 (Limitations and Future Works).``
>
> [1] Y. Xiang, T. Schmidt, V. Narayanan, and D. Fox. Posecnn: A convolutional neural network for 6d object pose estimation in cluttered scenes. arXiv preprint arXiv:1711.00199, 2017.
>
> [2] C. Xie, Y. Xiang, A. Mousavian, and D. Fox. Unseen object instance segmentation for robotic environments. IEEE Transactions on Robotics, 37(5):1343–1359.
>
> __Q2.__ The limitations of superquadrics as a shape representation are not discussed.
>
> __A2.__ Thank you for raising this point. ``We also have added the commentary on the limitations of using superquadrics to Section 4.3 (Limitations and Future Works).`` Since SQPD-Net considers single superquadric-shaped objects, it is not trivial to apply it directly to more complex or non-convex shapes. As research on representing objects in multiple superquadrics progress [3,4], extending our approach to multiple superquadric-shaped objects remains future work.
>
> [3] D. Paschalidou, A. O. Ulusoy, and A. Geiger. Superquadrics revisited: Learning 3d shape parsing beyond cuboids. In Proceedings of the IEEE/CVF Conference on Computer Vision and Pattern Recognition, pages 10344–10353, 2019
>
> [4] W. Liu, Y. Wu, S. Ruan, and G. S. Chirikjian. Robust and accurate superquadric recovery: a probabilistic approach. In Proceedings of the IEEE/CVF Conference on Computer Vision and Pattern Recognition, pages 2676–2685, 2022.

---

> ### Author Response · Authors · 2022-08-23
> **Response to Reviewer tXbi (Revised Paper and Supplementary Material)**
>
> **Comment:**
>
> Thank you very much for your constructive feedback. In response to the many constructive suggestions we have received, we have spent the past week revising our manuscript accordingly. __The revised paper and supplementary materials are attached to this comment.__ In an attempt to answer the reviewer questions, and to better clarify and validate our contributions, we include the new additional contents in the revised manuscript as follows:
>
> * we have added Appendix E.3 (Pushing Dynamics Learning on Real-world Pushing Data), which validates the motivation and significance of our method more clearly;
> * we have added Appendix E.4 (Pushing Manipulation via Interaction);
> * we have added Section 4.3 (Limitations and Future Works) to describe the limitations of our approach more clearly;
> * we have added Appendix A.4 (Invariance and Equivariance in Deep Learning and Robot Manipulations);
> * we have added pushing manipulation demo videos in the following Youtube link: https://youtu.be/OLoAHhf7vk0;
> * we have added some more details of pushing manipulation methods in Appendix D;
> * we have fixed some typos and clarified some missing definitions.
>
> In addition to these changes, we have noted that there was a minor glitch in our data generation method; the robot exhibited irregular movements on the simulator when the robot's inverse kinematics were not solved properly, resulting in data pushing the objects irregularly. It occurred a few times during the experiment (15 data in a total of 14400 data). We have generated a new dataset and trained our model and all of the baseline models on this new dataset, then we have updated our figures and table accordingly; specifically, Tables 2, 3, Figures 8 and 10 of the manuscript, and Figures 13 and 14 of the appendix (of the revised version) are updated. These irregular movements were rare, thus none of our observations and conclusions have changed. We regret that it was a mistake that should have been carefully checked prior to submission. After rigorous verifications, we are now very confident that the results in the updated manuscript are genuine.
>
> Below we provide detailed responses to each of the reviewer comments. When referencing any major changes and the addition of new content to the revised manuscript, we have indicated those passages ``like this``.
>
> **Zip File:**
>
> /attachment/de176e62d0041457bc8fef218a6bea244543a915.zip

---

### Official Review · Reviewer_T36J · 2022-07-31

**Originality:** Good
**Technical Quality:** Very Good
**Clarity Of Presentation:** Very Good
**Impact:** 3

**Recommendation:**

Weak Accept: I recommend accepting the paper, but will not argue for my recommendation if the majority of other reviewers have a different opinion.

**Summary:**

The main contribution of this paper is the design of an SE(2)-equivariant pushing dynamics model, which is then used in a model-based learning method. Objects are first detected by fitting superquadrics onto depth images and/or point cloud representations, which is an efficient way to improve pose definitions while simultaneously being a simpler description mechanism than the raw visual observation. Then through a series of transformations, they are concatenated and max pooled to enable equivariance, and input into a predictor model that predicts next poses. This model is then used by sampling based MPC methods to learn the sequence of actions needed to achieve the goal.

**Issues:**

- Some variable definitions seemed to be missing in the equations.
- A brief explanation of why only simple objects were considered in this paper, or some experiments with more complex objects would strengthen the contributions of the paper.
- Clarification on whether open-loop or closed-loop action sampling is used.


**Quality Of The Limitations Section:**

Limitations are addressed clearly

**Reviewer Expertise:**

3: The reviewer is fairly confident that the evaluation is correct

**Robotics Focus:**

Sufficient demonstration on hardware

**Strengths And Weaknesses:**

Strengths:
- The proposed method seems solid, easy to understand and improves upon the baselines in the paper. The equivariance property is useful for robot manipulation and it was interesting to see how this was encapsulated in the method.
- The use of superquadrics is simplify object representations and improve pose descriptions is elegant.
- The experiments are thorough, with successful sim-to-real transfer. Additionally, the robot learning to push flat objects to the edge of the table in order to grasp them is a good indication of the efficacy of the proposed method.
- The paper is well-written, with proper discussion, good use of figures and clear explanations.

Weaknesses:
- The shapes of objects used seemed to be a bit simple (cylinders and cuboids). Given the use of superquadrics, it would have been great to see some complex shapes, or even a commentary on limitations if unable to.
- Perhaps not directly a weakness of the paper, but it would have been great to see some videos of the experiments.

Improvements:
- A sentence or two explaining how long the training process takes, and a clarification on whether the actions were executed in an open loop or closed-loop manner would be good (are the actions and candidate grasp poses sampled once or resampled every so often?).

**Summary Of Recommendation:**

The proposed method seems good, both in terms of the logic of the design and the experimental results, though I can’t comment on the novelty. The paper is well-written, with extremely detailed description of the method, thorough experiments and clear discussion of the results. The back and forth between the main paper and the supplementary Appendix (especially the related works section being in the Appendix) made it a bit frustrating to read, but that’s just a minor issue. I also have not verified all the derivations in the paper, but there seemed to be occasional missing definitions of variables. The authors took a simple (yet important) observation of equivariance in learning dynamics and used it well to improve robot manipulation. In my opinion, overall, it is a solid paper.

---

> ### Author Response · Authors · 2022-08-23
> **Response to Reviewer T36J (1/1)**
>
> __Q1.__ The shapes of objects used seemed to be a bit simple (cylinders and cuboids). Given the use of superquadrics, it would have been great to see some complex shapes, or even a commentary on limitations if unable to.
>
> __A1.__ Thank you for this suggestion. ``We have added the commentary on the limitations of using superquadrics to Section 4.3 (Limitations and Future Works).`` Since SQPD-Net considers single superquadric-shaped objects, it is not trivial to apply it directly to more complex or non-convex shapes. As research on representing objects in multiple superquadrics progress [1,2], extending our approach to multiple superquadric-shaped objects remains future work.
>
> [1] D. Paschalidou, A. O. Ulusoy, and A. Geiger. Superquadrics revisited: Learning 3d shape parsing beyond cuboids. In Proceedings of the IEEE/CVF Conference on Computer Vision and Pattern Recognition, pages 10344–10353, 2019
>
> [2] W. Liu, Y. Wu, S. Ruan, and G. S. Chirikjian. Robust and accurate superquadric recovery: a probabilistic approach. In Proceedings of the IEEE/CVF Conference on Computer Vision and Pattern Recognition, pages 2676–2685, 2022.
>
> __Q2.__ Perhaps not directly a weakness of the paper, but it would have been great to see some videos of the experiments.
>
> __A2.__ ``We have added some real-world pushing manipulation demo videos in the following Youtube link`` (https://youtu.be/OLoAHhf7vk0) to provide more information. We hope this is some help.
>
> __Q3.__ A sentence or two explaining how long the training process takes, and a clarification on whether the actions were executed in an open loop or closed-loop manner would be good (are the actions and candidate grasp poses sampled once or resampled every so often?).
>
> __A3.__ Thank you for the suggestion. ``We have reported the time the training process takes in Appendix D.2 (Details for Pushing Dynamics Learning Experiments).`` The segmentation network, recognition network, and motion prediction network (SQPD-Net) take approximately 21.5 hours, 34 hours, and 16 hours to train on the RTX3080 Ti, respectively. Also, ``We have added a clarification on whether the actions were executed in an open loop or closed-loop manner in Appendix D.3 (Details for Pushing Manipulation Experiments).`` Since we use Model Predictive Control (MPC) which solves the optimal control problem at every timestep, the optimal actions are newly calculated at every timestep in a closed-loop manner. The candidate grasp poses for the target object are also resampled every timestep.
>
> __Q4.__ Some variable definitions seemed to be missing in the equations.
>
> __A4.__ Thank you for pointing it out. In the revised paper, ``we have reviewed the entire manuscript rigorously and clarified some missing definitions.``

---

> ### Author Response · Authors · 2022-08-23
> **Response to Reviewer T36J (Revised Paper and Supplementary Material)**
>
> **Comment:**
>
> Thank you very much for your constructive feedback. In response to the many constructive suggestions we have received, we have spent the past week revising our manuscript accordingly. __The revised paper and supplementary materials are attached to this comment.__ In an attempt to answer the reviewer questions, and to better clarify and validate our contributions, we include the new additional contents in the revised manuscript as follows:
>
> * we have added Appendix E.3 (Pushing Dynamics Learning on Real-world Pushing Data), which validates the motivation and significance of our method more clearly;
> * we have added Appendix E.4 (Pushing Manipulation via Interaction);
> * we have added Section 4.3 (Limitations and Future Works) to describe the limitations of our approach more clearly;
> * we have added Appendix A.4 (Invariance and Equivariance in Deep Learning and Robot Manipulations);
> * we have added pushing manipulation demo videos in the following Youtube link: https://youtu.be/OLoAHhf7vk0;
> * we have added some more details of pushing manipulation methods in Appendix D;
> * we have fixed some typos and clarified some missing definitions.
>
> In addition to these changes, we have noted that there was a minor glitch in our data generation method; the robot exhibited irregular movements on the simulator when the robot's inverse kinematics were not solved properly, resulting in data pushing the objects irregularly. It occurred a few times during the experiment (15 data in a total of 14400 data). We have generated a new dataset and trained our model and all of the baseline models on this new dataset, then we have updated our figures and table accordingly; specifically, Tables 2, 3, Figures 8 and 10 of the manuscript, and Figures 13 and 14 of the appendix (of the revised version) are updated. These irregular movements were rare, thus none of our observations and conclusions have changed. We regret that it was a mistake that should have been carefully checked prior to submission. After rigorous verifications, we are now very confident that the results in the updated manuscript are genuine.
>
> Below we provide detailed responses to each of the reviewer comments. When referencing any major changes and the addition of new content to the revised manuscript, we have indicated those passages ``like this``.
>
> **Zip File:**
>
> /attachment/f29542c371c166ebd599035ca244eadaac44367a.zip

---

### Official Review · Reviewer_uaxP · 2022-08-01

**Originality:** Fair
**Technical Quality:** Fair
**Clarity Of Presentation:** Good
**Impact:** 2

**Recommendation:**

Weak Accept: I recommend accepting the paper, but will not argue for my recommendation if the majority of other reviewers have a different opinion.

**Summary:**

This paper presents SuperQuadric Pushing Dynamics Network (SQPD-Net), a learning-based simulator for modeling the interactions between objects (represented as superquadrics) on a planar surface. This work is motivated by the observation that if the whole pushing configuration is transformed by a planar rigid-body transformation, then the pushing result should also be the result of the same transformation. The authors thus propose to incorporate SE(2)-equivariance in the learning framework by considering an object-centric coordinate frame when modeling the object-object interactions.

The authors have compared the proposed method with a few existing learning-based multi-object interaction simulators and showed improved prediction accuracy thanks to the built-in SE(2)-equivariance. They have used the learned model together with model-based planning algorithms for three types of downstream manipulation tasks (i.e., moving, singulation, and grasping) both in simulation and the real world.

**Issues:**

Concrete and comprehensive comparisons with physics-based models could be necessary to show the significance of this paper. Depending on the objective of this work, different types of experiments could be needed to justify the necessity of learning in the specific scenarios considered in this paper.

I would like to see how well the model work in more complicated scenarios that involve multi-object interactions (e.g., pushing a chain or a pile of objects). Currently, the real-world experiments mostly involve pushing just one object at each manipulation step.

It would help the readers better understand the difficulty of the tasks and the performance of the method by showing video demonstrations of the task completion processes.

Please see the weaknesses section for more questions and issues.

**Quality Of The Limitations Section:**

Additional details required

**Reviewer Expertise:**

5: The reviewer is absolutely certain that the evaluation is correct and very familiar with the relevant literature

**Robotics Focus:**

Sufficient demonstration on hardware

**Strengths And Weaknesses:**

**[Strengths]**

This paper is tackling an important problem of how to add priors and invariances in the learning-based simulators to improve the prediction accuracy and long-term rollout stability.

The idea of incorporating SE(2)-equivariance in the planar pushing tasks nicely captures the structure of the underlying system and has shown improved performance than the alternatives that do not consider such structure.

It is nice that the authors have shown downstream planning results using the learned model both in simulation and in the real world.

**[Weaknesses]**

While I like the general direction of this paper, I have concerns regarding the motivation, the evaluation, and the lack of comparison with more necessary baselines.

I do not feel that the paper has sufficiently motivated why we want to learn the rigid-body planar pushing simulator from data. We have very good physics-based rigid body simulators. Even the proposed model itself is learned from a rigid body simulator (i.e., Pybullet) and evaluated by using Pybullet as the ground truth. If we add Pybullet as a baseline for evaluating the prediction accuracy, the error would be zero. Why don't we directly use Pybullet (or other rigid-body simulators) for real-world experiments? What value do the learning-based dynamics models bring in the specific evaluating scenario?

I could think of the following potential reasons.
- Will it be easier for the learned model to perform online adaptation given real-world observation (e.g., identify the center of mass, friction, distribution of supporting force) --- in other words, would it be more accurate in fitting the real-world observation data? If this is the case, more system identification or online adaptation experiments compared with physics-based simulators are needed for the real-world experiments.
- Will the learned model be able to simulate systems that are otherwise hard for physics-based models to simulate? This could be true if the authors are dealing with objects with more complicated physical properties, but we have a very good understanding of rigid body dynamics. It is hard for me to imagine, in rigid-body planar pushing scenarios, how the learning-based method is better than the physics-based simulators, which already have the priors built in (e.g., SE(2)-equivariance, objects stay rigid).
- The learning-based method can potentially learn models from observation data without the assumption of access to full-state information. However, in this paper, the authors assume a perception model that can produce the full state information of the environment (e.g., object shape and pose), which could also be the inputs to a physics-based simulator to predict the future.
- Can the learned model provide better optimization results for downstream planning tasks? If this is the case, the authors need to show that the learning-based method is more effective at accomplishing the manipulation tasks than using the physics-based simulators (e.g., use the same sampling-based MPCs with Pybullet).

Depending on the authors' motivation, more extensive experiments and comparisons with physics-based simulators are needed to justify the significance of this paper.

The real-world experiments are simple. It seems that only one of the objects is moving during the majority (or all?) of the task completion process, which only constitutes a very small portion of what can happen in the scenario. How well does the method work if there are more complicated multi-object interactions (e.g., the pushed object also pushes other objects, pushing objects in a pile)?

I'm also curious about situations where the SE(2)-equivariance does not apply. For example, what if the friction coefficients are different in different regions of the table? What if the table is not perfectly flat and the distribution of the supporting force is different in different areas?

There are also a lot of works that use graph neural networks that capture the compositionality of the underlying systems and model multi-object dynamics (e.g., [1] and many follow-up works). Therefore, comparisons with these works could also be essential to show the significance of the proposed method, demonstrate the advantages of the incorporated priors on SE(2)-equivariance, and expand its impact.

[1] Peter W. Battaglia, Razvan Pascanu, Matthew Lai, Danilo Rezende, Koray Kavukcuoglu, "Interaction Networks for Learning about Objects, Relations and Physics"

**Summary Of Recommendation:**

While I like the general direction of this paper, I have concerns regarding the motivation, the evaluation, and the lack of comparison with more necessary baselines (especially the physics-based simulators like Pybullet).

=============

**[Post-rebuttal recommendation]**

I have read the rebuttal and the reviews from other reviewers. I appreciate the authors' effort in providing new experimental results in response to my questions, especially the comparison with PyBullet by learning models directly from real-world observations. As a result, I updated my score from Weak Reject to Weak Accept.

---

> ### Author Response · Authors · 2022-08-23
> **Response to Reviewer uaxP (2/2)**
>
> __Q3.__ I'm also curious about situations where the SE(2)-equivariance does not apply. For example, what if the friction coefficients are different in different regions of the table? What if the table is not perfectly flat and the distribution of the supporting force is different in different areas?
>
> __A3.__ Thank you for pointing out these issues. When the friction coefficients are different in different regions of the table, a locally SE(2)-equivariant model -- SE(2) space is divided into several subspaces and the model is equivariant only within each subspace -- can be considered. If the table is not perfectly flat, we may define another equivariant dynamics model corresponding to the symmetry of the table.
>
> The other example is the case the distribution of the supporting force is different in different areas, especially the case when pushing an object with non-uniform mass distribution. First of all, although it may already be apparent to the reviewer, we want to clarify that our model aims to guarantee SE(2)-equivariance in the situation where objects are placed in a different pose transformed by SE(2) transformation, but not aims to design equivariant models that capture the symmetry of the objects. In other words, our equivariant model considers the cases in which the objects and pushing actions are transformed by the same SE(2) transformation, not the cases in which the action pushes another symmetrical part of the object. That being said, if we can consistently predict the reference poses of the objects (e.g., pre-specified poses in CAD models), our SE(2)-equivariant model is applicable regardless of the mass distribution. Of course, predicting reference poses is not easy with only depth images, so this is still one of the limitations of our approach. In this case, we have to utilize additional information such as RGB images [1,2]. ``We have added these discussions about these limitations to Section 4.3 (Limitations and Future Works).``
>
> [1] Y. Xiang, T. Schmidt, V. Narayanan, and D. Fox. Posecnn: A convolutional neural network for 6d object pose estimation in cluttered scenes. arXiv preprint arXiv:1711.00199, 2017.
>
> [2] C. Xie, Y. Xiang, A. Mousavian, and D. Fox. Unseen object instance segmentation for robotic environments. IEEE Transactions on Robotics, 37(5):1343–1359.
>
> __Q4.__ There are also a lot of works that use graph neural networks that capture the compositionality of the underlying systems and model multi-object dynamics (e.g., [1] and many follow-up works). Therefore, comparisons with these works could also be essential to show the significance of the proposed method, demonstrate the advantages of the incorporated priors on SE(2)-equivariance, and expand its impact.
>
> __A4.__ Thank you for raising this point. Building a pushing dynamics model using a graph neural network is a very interesting idea. The suggested existing works are rather orthogonal to our work which is complementary to the proposed SE(2)-equivariance principle and hence we do not consider them as subjects to direct comparisons. Nevertheless, testing and expanding our model with graph neural networks and comparing the results could be definitely an interesting future research direction.
>
> __Q5.__ It would help the readers better understand the difficulty of the tasks and the performance of the method by showing video demonstrations of the task completion processes.
>
> __A5.__ ``We have added some real-world pushing manipulation demo videos in the following Youtube link`` (https://youtu.be/OLoAHhf7vk0) to provide more information. We hope this is some help.

---

> ### Author Response · Authors · 2022-08-23
> **Response to Reviewer uaxP (1/2)**
>
> __Q1.__ I do not feel that the paper has sufficiently motivated why we want to learn the rigid-body planar pushing simulator from data. We have very good physics-based rigid body simulators. Even the proposed model itself is learned from a rigid body simulator (i.e., Pybullet) and evaluated by using Pybullet as the ground truth. If we add Pybullet as a baseline for evaluating the prediction accuracy, the error would be zero. Why don't we directly use Pybullet (or other rigid-body simulators) for real-world experiments? What value do the learning-based dynamics models bring in the specific evaluating scenario? Depending on the authors' motivation, more extensive experiments and comparisons with physics-based simulators are needed to justify the significance of this paper.
>
> __A1.__
> Thank you for raising this issue. We agree with your comment that the motivation why we have to learn the rigid-body planar pushing simulator from data was insufficient. The motivation of learning is that a physics-based simulator that precisely models the physical interactions cannot be used since we are given unseen objects with only vision data (physical quantities such as friction coefficient are not available). Currently, we learn the dynamics model from simulation and apply it to real-world manipulation, so it is not sufficient to verify the motivation as the reviewer notes. To justify this motivation, we should train our model on a real-world dataset and compare its performance with a physics-based simulator. To validate the motivation and significance of our method more clearly in this context, ``we have added Appendix E.3 (Pushing Dynamics Learning on Real-world Pushing Data).``
>
> In this added Appendix, we compare the motion prediction accuracy between our model trained on a real-world dataset and a physics-based simulator. Collecting real-world data is difficult compared to simulation; for ground-truth annotation, we set up an environment that can estimate the pose of the object from the point cloud observation. After collecting ground-truth pushing data in the real-world, we train our model and compare the prediction performance of our trained model on real-world dataset with the physics-based simulator (PyBullet). We have confirmed that __our approach outperforms the PyBullet simulator qualitatively and quantitatively,__ and especially, our model performs much better in terms of prediction of the objects' orientations. We refer to Appendix E.3 for the additional details. We hope that this additional section better justifies the significance of our paper.
>
> __Q2.__ The real-world experiments are simple. It seems that only one of the objects is moving during the majority (or all?) of the task completion process, which only constitutes a very small portion of what can happen in the scenario. How well does the method work if there are more complicated multi-object interactions (e.g., the pushed object also pushes other objects, pushing objects in a pile)?
>
> __A2.__
> We appreciate the reviewer’s question on “how well the method works if there are more complicated multi-object interactions.” To verify that, ``we have added some pushing manipulation demo videos in the following Youtube link: ``https://youtu.be/OLoAHhf7vk0. The videos contain some examples of pushing manipulation that the interaction occurs; especially, in the task of grasping in a cluttered environment, it is often seen that the task is successfully performed by utilizing multi-objects interaction.
>
> Additionally, ``we have defined a new task that essentially requires the interaction between multi-objects, and have seen how our method works in this case in Appendix E.4 (Pushing Manipulation via Interaction).`` Briefly, we define a situation where the target object should be moved to a specified position, but the target object cannot be pushed. The robot must move the target object to the desired position by pushing the other objects; this essentially requires interaction. We have confirmed that our method also works successfully in the situations that require interaction.

---

> ### Author Response · Authors · 2022-08-23
> **Response to Reviewer uaxP (Revised Paper and Supplementary Material)**
>
> **Comment:**
>
> Thank you very much for your constructive feedback. In response to the many constructive suggestions we have received, we have spent the past week revising our manuscript accordingly. __The revised paper and supplementary materials are attached to this comment.__ In an attempt to answer the reviewer questions, and to better clarify and validate our contributions, we include the new additional contents in the revised manuscript as follows:
>
> * we have added Appendix E.3 (Pushing Dynamics Learning on Real-world Pushing Data), which validates the motivation and significance of our method more clearly;
> * we have added Appendix E.4 (Pushing Manipulation via Interaction);
> * we have added Section 4.3 (Limitations and Future Works) to describe the limitations of our approach more clearly;
> * we have added Appendix A.4 (Invariance and Equivariance in Deep Learning and Robot Manipulations);
> * we have added pushing manipulation demo videos in the following Youtube link: https://youtu.be/OLoAHhf7vk0;
> * we have added some more details of pushing manipulation methods in Appendix D;
> * we have fixed some typos and clarified some missing definitions.
>
> In addition to these changes, we have noted that there was a minor glitch in our data generation method; the robot exhibited irregular movements on the simulator when the robot's inverse kinematics were not solved properly, resulting in data pushing the objects irregularly. It occurred a few times during the experiment (15 data in a total of 14400 data). We have generated a new dataset and trained our model and all of the baseline models on this new dataset, then we have updated our figures and table accordingly; specifically, Tables 2, 3, Figures 8 and 10 of the manuscript, and Figures 13 and 14 of the appendix (of the revised version) are updated. These irregular movements were rare, thus none of our observations and conclusions have changed. We regret that it was a mistake that should have been carefully checked prior to submission. After rigorous verifications, we are now very confident that the results in the updated manuscript are genuine.
>
> Below we provide detailed responses to each of the reviewer comments. When referencing any major changes and the addition of new content to the revised manuscript, we have indicated those passages ``like this``.
>
>
> **Zip File:**
>
> /attachment/a4ed82647ecbec9b86464aafb87f8c672ad89496.zip

---

### Author Response · Authors · 2022-08-23
**Response to All Reviewers**

**Comment:**

Thank you very much for your constructive feedback. In response to the many constructive suggestions we have received, we have spent the past week revising our manuscript accordingly. __The revised paper and supplementary materials are attached to this comment.__ In an attempt to answer the reviewer questions, and to better clarify and validate our contributions, we include the new additional contents in the revised manuscript as follows:

* we have added Appendix E.3 (Pushing Dynamics Learning on Real-world Pushing Data), which validates the motivation and significance of our method more clearly;
* we have added Appendix E.4 (Pushing Manipulation via Interaction);
* we have added Section 4.3 (Limitations and Future Works) to describe the limitations of our approach more clearly;
* we have added Appendix A.4 (Invariance and Equivariance in Deep Learning and Robot Manipulations);
* we have added pushing manipulation demo videos in the following Youtube link: https://youtu.be/OLoAHhf7vk0;
* we have added some more details of pushing manipulation methods in Appendix D;
* we have fixed some typos and clarified some missing definitions.

In addition to these changes, we have noted that there was a minor glitch in our data generation method; the robot exhibited irregular movements on the simulator when the robot's inverse kinematics were not solved properly, resulting in data pushing the objects irregularly. It occurred a few times during the experiment (15 data in a total of 14400 data). We have generated a new dataset and trained our model and all of the baseline models on this new dataset, then we have updated our figures and table accordingly; specifically, Tables 2, 3, Figures 8 and 10 of the manuscript, and Figures 13 and 14 of the appendix (of the revised version) are updated. These irregular movements were rare, thus none of our observations and conclusions have changed. We regret that it was a mistake that should have been carefully checked prior to submission. After rigorous verifications, we are now very confident that the results in the updated manuscript are genuine.

Below we provide detailed responses to each of the reviewer comments. When referencing any major changes and the addition of new content to the revised manuscript, we have indicated those passages ``like this``.

**Zip File:**

/attachment/f7f051aaf5b204a19392520d447da656d7f8d61d.zip

---

### Meta-Review · Area_Chair_g9MP · 2022-08-02

**Recommendation:** Accept (Oral)
**Confidence:** 4

**Metareview:**

Strengths:

+ The paper tackles the use of equivariance in learning forward models, which the reviewers agree is a very important topic highly relevant to manipulation.
+ Results show significant improvement compared to baselines.
+ The paper is generally clear and well written (although occasional variable definitions are missing and some notations could be changed for clarity)

Weaknesses:
- While reviewers generally appreciated the use of superquadrics to approximate object shapes, this approach also raises very important questions. For simple objects that can be easily fitted using a single superquadric, the field has good analytical pushing models. What is the advantage of learning, especially since it has to be done through a simulator? Note that this aspect was addressed in the discussion stage and a clear point on on-robot learning was presented.
- How would this method apply to more complex or non-convex shapes? While this is not a topic easily addressed in the current version, the authors present a more clear discussion of it and possibilities for future work.
- Analytical models also have the ability to consider complex phenomena that vision can not perceive, such as different responses due to different pressure distributions of the contact area. In such cases, an incorrect equivariance assumption might in fact be detrimental. This aspect is not discussed.
- The paper does not discuss highly relevant recent literature that makes use of equivariance in the context of representation or reinforcement learning. This point has been addressed in the discussion and revision.


**Best Paper Nomination:**

No

---

> ### Author Response · Authors · 2022-08-23
> **Response to Meta Review (2/2)**
>
> __Q4.__ The paper does not discuss highly relevant recent literature that makes use of equivariance in the context of representation or reinforcement learning.
>
> __A4.__
> We fully agree with the reviewer's comment that the review of related studies about invariance and equivariance in deep learning and robot manipulations was missing. We have accordingly ``added the related work section about invariance and equivariance in Appendix A.4 (Invariance and Equivariance in Deep Learning and Robot Manipulations).`` In this Appendix, we compare our work with these related works and clearly state at which points the proposed method is novel.

---

> ### Author Response · Authors · 2022-08-23
> **Response to Meta Review (1/2)**
>
> __Q1.__ While reviewers generally appreciated the use of superquadrics to approximate object shapes, this approach also raises very important questions. For simple objects that can be easily fitted using a single superquadric, the field has good analytical pushing models. What is the advantage of learning, especially since it has to be done through a simulator?
>
> __A1.__ Thank you for raising this issue. We agree with the reviewer's comment that the advantage of learning for pushing dynamics should be more emphasized. The motivation of learning is that a physics-based simulator that precisely models the physical interactions cannot be used since we are given unseen objects with only vision data (quantities such as objects' shapes and mass and friction coefficient are not available). Currently, we learn the dynamics model from simulation and apply it to real-world manipulation, so it is not sufficient to verify the motivation as the reviewer notes. To justify this motivation, we should train our model on a real-world dataset and compare its performance with a physics-based simulator. To validate the motivation and significance of our method more clearly in this context, ``we have added Appendix E.3 (Pushing Dynamics Learning on Real-world Pushing Data).``
>
> In this added Appendix, we compare the motion prediction accuracy between our model trained on a real-world dataset and a physics-based simulator. Collecting real-world data is difficult compared to simulation; for ground-truth annotation, we set up an environment that can estimate the pose of the object from the point cloud observation. After collecting ground-truth pushing data in the real-world, we train our model and compare the prediction performance of our trained model on real-world dataset with the physics-based simulator (PyBullet). We have confirmed that \textit{our approach outperforms PyBullet simulator qualitatively and quantitatively,} and especially, our model performs much better in terms of prediction of the objects' orientations. We refer to Appendix E.3 for the additional details. We hope that this additional section better justifies the significance of our paper.
>
> __Q2.__ How would this method apply to more complex or non-convex shapes?
>
> __A2.__ Thank you for raising this issue. ``We have added the commentary on limitations of using superquadrics to Section 4.3 (Limitations and Future Works).`` Since SQPD-Net considers single superquadric-shaped objects, it is not trivial to apply it directly to more complex or non-convex shapes. As research on representing objects in multiple superquadrics progress [1,2], extending our approach to multiple superquadric-shaped objects remains future work.
>
> [1] D. Paschalidou, A. O. Ulusoy, and A. Geiger. Superquadrics revisited: Learning 3d shape parsing beyond cuboids. In Proceedings of the IEEE/CVF Conference on Computer Vision and Pattern Recognition, pages 10344–10353, 2019
>
> [2] W. Liu, Y. Wu, S. Ruan, and G. S. Chirikjian. Robust and accurate superquadric recovery: a probabilistic approach. In Proceedings of the IEEE/CVF Conference on Computer Vision and Pattern Recognition, pages 2676–2685, 2022.
>
> __Q3.__ Analytical models also have the ability to consider complex phenomena that vision can not perceive, such as different responses due to different pressure distributions of the contact area. In such cases, an incorrect equivariance assumption might in fact be detrimental. This aspect is not discussed.
>
> __A3.__ Thank you for pointing out this issue. We agree with the reviewer's comment that different mass distributions will lead to different motions (e.g., a cube with a shifted center of mass). First of all, although it may already be apparent to the reviewer, we want to clarify that our model aims to guarantee SE(2)-equivariance in the situation where objects are placed in a different pose transformed by SE(2) transformation, but not aims to design equivariant models that capture the symmetry of the objects. In other words, our equivariant model considers the cases in which the objects and pushing actions are transformed by the same SE(2) transformation, not the cases in which the action pushes another symmetrical part of the object. That being said, if we can consistently predict the reference poses of the objects (e.g., pre-specified poses in CAD models), our SE(2)-equivariant model is applicable regardless of the mass distribution. Of course, predicting reference poses is not easy with only depth images, so this is still one of the limitations of our approach. In this case, we have to utilize additional information such as RGB images [1,2]. ``We have added these discussions about these limitations to Section 4.3 (Limitations and Future Works).``

---

> ### Author Response · Authors · 2022-08-23
> **Response to Meta Review (Revised Paper and Supplementary Material)**
>
> **Comment:**
>
> Thank you very much for your constructive feedback. In response to the many constructive suggestions we have received, we have spent the past week revising our manuscript accordingly. __The revised paper and supplementary materials are attached to this comment.__ In an attempt to answer the reviewer questions, and to better clarify and validate our contributions, we include the new additional contents in the revised manuscript as follows:
>
> * we have added Appendix E.3 (Pushing Dynamics Learning on Real-world Pushing Data), which validates the motivation and significance of our method more clearly;
> * we have added Appendix E.4 (Pushing Manipulation via Interaction);
> * we have added Section 4.3 (Limitations and Future Works) to describe the limitations of our approach more clearly;
> * we have added Appendix A.4 (Invariance and Equivariance in Deep Learning and Robot Manipulations);
> * we have added pushing manipulation demo videos in the following Youtube link: https://youtu.be/OLoAHhf7vk0;
> * we have added some more details of pushing manipulation methods in Appendix D;
> * we have fixed some typos and clarified some missing definitions.
>
> In addition to these changes, we have noted that there was a minor glitch in our data generation method; the robot exhibited irregular movements on the simulator when the robot's inverse kinematics were not solved properly, resulting in data pushing the objects irregularly. It occurred a few times during the experiment (15 data in a total of 14400 data). We have generated a new dataset and trained our model and all of the baseline models on this new dataset, then we have updated our figures and table accordingly; specifically, Tables 2, 3, Figures 8 and 10 of the manuscript, and Figures 13 and 14 of the appendix (of the revised version) are updated. These irregular movements were rare, thus none of our observations and conclusions have changed. We regret that it was a mistake that should have been carefully checked prior to submission. After rigorous verifications, we are now very confident that the results in the updated manuscript are genuine.
>
> Below we provide detailed responses to each of the reviewer comments. When referencing any major changes and the addition of new content to the revised manuscript, we have indicated those passages ``like this``.
>
> **Zip File:**
>
> /attachment/aecdadb2697d4e9a10272e9668d7b45de9cc687b.zip